# Changes in the vitreous body after experimental vitreous hemorrhage in rabbit: An interdisciplinary study

**Pengcheng Zhang**[1,2], **Weijia Yan**[3], **Hong Yan**[1]*

1 Xi'an People's Hospital (Xi'an Fourth Hospital), Shaanxi Eye Hospital, Affiliated Xi'an Fourth Hospital, Northwestern Polytechnical University, Xi'an, Shaanxi, China, 2 Department of Ophthalmology, General Hospital of Central Theater Command, PLA, Wuhan, Hubei, China, 3 Department of Ophthalmology, University of Heidelberg, Heidelberg, Germany

* yan2128ts@med.nwu.edu.cn

**Data Availability Statement:** All relevant data are within the manuscript and its Supporting Information file.

**Funding:** This research was supported by National Nature Science Foundation of China (81873674,

## Abstract

### Purpose

To explore the changes in vitreous body after vitreous hemorrhage and assess its prognosis from the perspective of vitreoretinal interface.

### Methods

The experiment was performed on 32 New Zealand rabbits (64 eyes), weighing 2500–3000 g for 4 months and unlimited gender, which was injected with 0.2 mL of autologous blood into the center of vitreous cavity–the study group (right eyes), and the control one was treated in the same manner with equal volumes of saline. The rabbits were randomly and equally divided into the following four batches according to the days of observation: Days 3, 7, 14, and 30 after injection. IOP and severity grading were evaluated before rabbits' execution and eyeballs were enucleated. The anterior segment was separated to flow out the vitreous body naturally to detect the liquefaction degree and viscosity. Then, chemical composition of electrolytes, PCT and bFGF were determined by colorimetry and enzyme-linked immunosorbent assay (ELISA). Finally, the incidence of posterior vitreous detachment (PVD) was observed after vitreous sampled. The studies were double-blind.

### Results

After injection, the extent of vitreous opacity and coagulum size decreased over time. Both the degree of liquefaction and the length of tow differed significantly between two groups at different time points (all $p < 0.001$). The liquefaction degree in the study group rose obviously from the Day 14, which the viscosity declined significantly on the initial time. Biochemical markers fluctuated temporarily, except for basic fibroblast growth factor (bFGF), which continued to rise and was correlated with the liquefaction degree ($r = 0.658$, $p < 0.001$). Besides, the incidence of PVD increased from the 14th day ($p < 0.05$), and it was highly positively correlated with the number of macrophages ($r = 0.934$; $p < 0.001$).

82070947, to Hong Yan), Xi'an Fourth Hospital Research Incubation Fund (LH-6) and Xi'an Talent Program (XAYC200021). The financial sources had no role in study desigh, data collection and analysis, decision to publish, or preparation of manuscript.

**Competing interests:** The authors have declared that no competing interests exist.

## Conclusion

After vitreous hemorrhage, the changes of the vitreous body are relatively minor earlier (2–4 weeks), but irreversible later. Specifically, the degree of liquefaction increases with a decrease in viscosity, and the chemotaxis of macrophages and bFGF induce incomplete PVD.

## Introduction

The outbreak of Novel Coronavirus SARS-CoV-2 (COVID-19) has put humans' lives and health at risk throughout the world [1]. As an exposed organ, the protection of eye is crucial in decelerating the spread and prevalence of the virus. Vitreous body, as the main content, plays a vital role in supporting ocular tissues and maintaining intraocular homeostasis. And, the role of vitreous hemorrhage (VH) has become one of research Hot Topics in secondary eye diseases. In fact, it does not constitute a separate disease, but rather a transitional period of vitreoretinopathy that arises from the periphery, or the vessels penetrating into the vitreous body [2], and subsequently causes liquefaction, epiretinal membrane, and even tractional detachment of retina. Therefore, it is critical to eliminate the role that hemorrhage plays in vitreous changes.

Hemorrhage or drugs residues in vitreous gel make it difficult to clear or diffuse from the visual axis [3, 4]. Researchers have attempted to dissolve blood, induce vitreous structural changes, or directly remove it. So, predicting the properties of the vitreous can help design bionic vitreous substitutes [5], analyze the intravitreal transport of therapeutics, and understand the pathology. However, there is a relative lack of discussion in vitreous, especially in terms of biomechanics, which is the motivation of this study.

Then, what happens in the vitreous body itself after VH and how it affects its course? The purpose of the current study is to investigate series of changes in vitreous after experimental VH and assess its prognosis from the perspective of vitreoretinal interface. Rabbit was selected as research animal for the following reasons: rabbit studies in ophthalmology are relatively mature, the composition of vitreous body of rabbit is similar to that of human, and the rabbit are gentle, easier to obtain and raise compared with other animals [6–8]. The blood absorption, an extracellular hemolysis, leads to the biomechanical fluctuation and redistribution of biochemical parameters of vitreous humor [9–12]. Furthermore, an endotoxin-induced inflammation-related protein called procalcitonin (PCT) and basic fibroblast growth factor (bFGF) are possible culprits [13].

## Material and methods

### Rabbits

The study protocol was approved by the Institutional Review Board of Xi'an People's Hospital (Xi'an Fourth Hospital) (No. 20220019), and performed in accordance with the Association for Research in Vision and Ophthalmology (ARVO) Statement for the Use of Animals in Ophthalmic and Vision Research.

The liquefaction degree and viscosity were used as main evaluation indexes, and the sample size was estimated base on t test data. Power package in the R programming language was utilized to analyze the sample size required in the current study, and a large effect size $d = 0.80$ was preset [14], with a statistical test power $1-\beta = 0.8$, significance level $\alpha = 0.05$. The estimated

minimum sample size was 26 subjects (eyes) in each group, and the estimated shedding rate was 10 percent. Finally, 32 New Zealand ordinary white rabbits (64 eyes, Experimental Animal Center, Air Force Medical University), 2.5–3.0 kg for 4 months and unlimited gender, were included, while any one with obvious lesions in the anterior segment or fundus of eye was excluded. They were housed singly, with food and water intake freely, strict temperature (23 ±2°C), humidity (50±5%) control, indoor ventilation, and adapted for at least 1 week before injection. Levofloxacin eye drops were put in to prevent infection (1/time, 3/day for 3 days). At the same time, feeding environment, as well as the health of rabbits, was monitored 3 times a day.

## Animal model construction

The rabbit was anesthetized with sodium pentobarbital (25–30 mg/kg) intramuscularly. The eyes were assigned into two groups: study group (right eyes)–full autologous blood injected, and control one (left eyes, evaluate possible adverse effects of ocular trauma or exogenous fluid)–injected with equal volumes of saline in the same manner [15, 16]. After collection from the marginal ear vein, 0.2 mL of autologous blood was injected into the right eye' center of vitreous cavity immediately, via pars plana approach to the upper temporal quadrant, 5 mm away from the limbus, following a standardized procedure. Then, an anterior chamber puncture was performed to soft eyes. All procedures were performed by the same surgeon with a 19-gauge needle under the control of an ophthalmoscope.

## Study design

The rabbits were randomly and evenly divided into the following four batches by Excel random number method according to the days of observation: Days 3, 7, 14, and 30 (n = 8 each time point) [15] after injection. First, IOP and severity grading [17] were evaluated before analysis of the vitreous properties. Then, they were executed by causing gas embolism and the eyeballs were enucleated using standard technique. The anterior segment was separated to expose and flow out the vitreous body naturally to petri dishes with a filter to detect the liquefaction degree and viscosity. 0.5-mL of vitreous humor was extracted using a micropipette (Eppendorf, 100–1000 μL) to a 1.5-mL tapered EP tube to determine the chemical composition of electrolytes, PCT and bFGF by colorimetry and enzyme-linked immunosorbent assay (ELISA) [15]. Finally, the eyeball was cut in half after vitreous sampled and the incidence of PVD was observed with scanning electron microscope (SEM) and transmission electron microscope (TEM), respectively. The treatments and measurements were double-blind, unaware to both operators and recorders, only to the researchers. The cage location did not differ in terms of lighting, temperature and noise, and there was no accidental loss of rabbits.

## Biomechanical, biochemical researches and ultrastructure observation on vitreous body

Vitreous humor was poured onto a pre-weighed fiber filter (12 cm ×15 cm, with mesh diameter of 1.5 mm), which was placed on a pre-weighed dish to separate the gel (Mgel) from the liquid (Mliq). The percentage of vitreous liquefaction was then calculated as follows: Mliq/(Mgel + Mliq). Next, the gel was lifted vertically at a constant speed of 0.5 mm/s with a 3 mm diameter glass applicator, which fixed to a speed control device and, the viscosity was graded on the basis of the length of its tow recorded when it happens to break: high, > 10mm; moderate, 5–10 mm; low, < 5 mm.

The remaining vitreous humor was extracted, centrifuged at 5000 rpm for 10 min, and the supernatant was quickly frozen at −20°C. Sample was uniformly thawed at room temperature,

and standard and tested antibodies were added to the antibody-coated microporous layer in turn according to time, dosage, order, and then washed thoroughly after warm incubation. The absorbance (OD value) was measured at 450 nm in a microplate reader to calculate the mass concentrations of PCT and bFGF.

The eyeballs cured with 3% glutaraldehyde were cut frontal, while those frozen in acetone were incised sagittal vertically to preserve vitreous changes.

## Statistical analysis

SPSS 22.0 (IBM Corporation, USA) was utilized for statistical analysis. P-P plot and Shapiro-Wilk test were used for normality test, and Levene in exploratory analysis was used for homogeneity of variance test. When the assumptions were not met, nonparametric test will be considered. For data analysis, the following statistical methods were used: average, standard deviation, ANOVA, $t$-test, proportions, chi-square test, Gamma test, pearson or spearman for correlation analysis, and confidence interval for $p = 0.95$.

## Results

### Course of VH

Compared with control group, the IOP in study one decreased on day 3 after injection, and then returned to baseline. However, the difference of IOP between two groups was not statistically significant ($10.68\pm1.58$ vs. $11.28\pm1.90$ mmHg, $t = -1.960$, $p = 0.059$; paired samples t test). Paired difference was ($-0.59\pm1.71$) mmHg, and 95% confidence interval of the difference was ($-1.21$, $0.02$) mmHg. The study group showed a progressive decreased in the grade of vitreous opacifications [17] from the initial grade-1 to grade-4 over time, which was segregated in five grades ophthalmoscopically, while the control one remained clear. As categoric variables among multiple groups with bidirectional order and different attributes, spearman analysis showed that there was a strong positive correlation between the opacities grade and postoperative follow-up time in the study group ($\rho = 0.860$, $p < 0.001$). Further, Gamma test was used to compare the numbers of eyeballs under different grades in each period of study group, and it showed significant difference ($G = 0.943$, $p < 0.001$), which means the extent of vitreous opacity decreased over time (Table 1). The average size of coagulum at different time intervals was 6.5, 5, 4, and 2 mm$^2$, respectively. The correlation of the coagulum size and follow-up time was highly significant via pearson test ($r = -0.976$, $p = 0.024$).

**Table 1. Eyeball number according to grading of vitreous opacities.**

| Grade | Study group | | | | Control group | | | |
|---|---|---|---|---|---|---|---|---|
| | Day-3 | Day-7 | Day-14 | Day-30 | Day-3 | Day-7 | Day-14 | Day-30 |
| 1 | 7 | 5 | | | | | | |
| 2 | 1 | 3 | 1 | | | | | |
| 3 | | | 5 | 2 | 1 | | | |
| 4 | | | 2 | 6 | 1 | 1 | | |
| 5 | | | | | 6 | 7 | 8 | 8 |

Grade 1 – completely opaque vitreous

Grade 2 – increased red reflex and no fundus detail visible

Grade 3 – patches of fundus visible between discrete vitreous opacities

Grade 4 – central vitreous clear with small residual opacities

Grade 5 – totally clean vitreous.

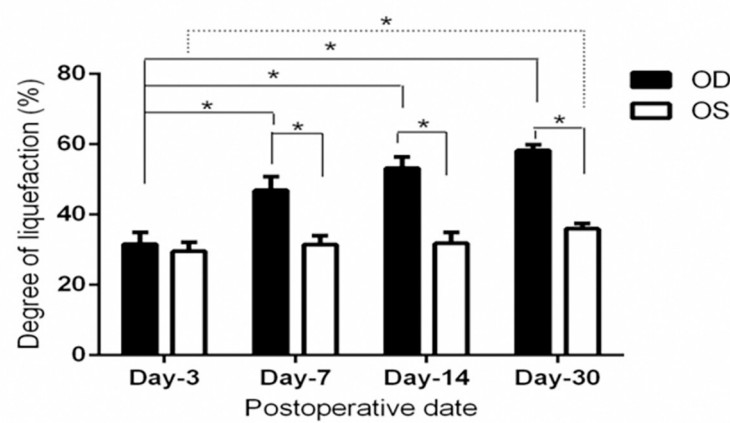

**Fig 1. Comparison of liquefaction degree of binocular vitreous during the course of VH (*p < 0.05).**

## Degree of liquefaction and viscosity

The difference of liquefaction degree between the paired two groups was statistically significant (48.56±10.88 vs. 32.09±3.16%, $t$ = 9.978, $p$ = 0.000). Paired difference was (16.46±9.33) %, and 95% confidence interval of the difference was (13.10, 19.83) %. There was no significant difference Day 3 after operation between two groups ($t$ = 1.826, $p$ = 0.111), but the difference in the remaining periods were statistically significant ($t$ = 9.066, 18.793, 29.439, all $p < 0.001$) (Fig 1 and Table 2). The data of liquefaction degree at different time points in the study group were in line with normal distribution but without equal variance, and the differences were not all the same, with a statistically significant ($H$ = 24.319, $p < 0.001$; kruskal-wallis H test). The mean rank of days 3, 7, 14, and 30 after injection were 4.50, 15.00, 19.38, and 27.12, respectively. Post-hoc pairwise comparison showed that there were statistically significant differences between day-3 and day-14 (adjusted $p$ = 0.009), and between day-3 and day-30 (adjusted $p < 0.001$), while there was no difference among the remaining points. It is noteworthy that the liquefaction degree in the controls also slightly increased ($F$ = 12.842, $p < 0.001$; one-way ANOVA, normal with equaled variance), and it was higher on day-30 than the remaining periods (all $p < 0.05$; pairwise). The viscosity of the vitreous humor in study group was graded moderate on day 3, and then became low, while it kept high in control one all the time. The difference of tow lengths between paired binocular gel was statistically significant (3.64±1.90 vs. 12.09±1.07 mm, $t$ = -26.802, $p < 0.001$), and the differences were obvious between two groups at each time point (all $p < 0.001$, Table 2).

**Table 2. Comparison of biomechanical characteristics of binocular vitreous at different time intervals (n = 8, $\bar{x}$ ±s).**

| Biomechanics | | Study group | Control group | $t$ | $p$ value |
|---|---|---|---|---|---|
| Degree of Liquefaction (%) | Day-3 | 31.57±3.36 | 29.52±2.56 | 1.826 | 0.111 |
| | Day-7 | 50.31±5.53 | 31.05±2.38 | 9.066 | 0.000 |
| | Day-14 | 54.40±2.77 | 31.88±2.03 | 18.793 | 0.000 |
| | Day-30 | 57.96±1.19 | 35.93±1.52 | 29.439 | 0.000 |
| Tow length of gel (mm) | Day-3 | 6.56±0.73 | 12.63±1.06 | -13.324 | 0.000 |
| | Day-7 | 3.25±0.89 | 12.25±0.85 | -20.785 | 0.000 |
| | Day-14 | 2.69±0.75 | 11.94±1.05 | -20.248 | 0.000 |
| | Day-30 | 2.06±0.56 | 11.56±1.21 | -20.158 | 0.000 |

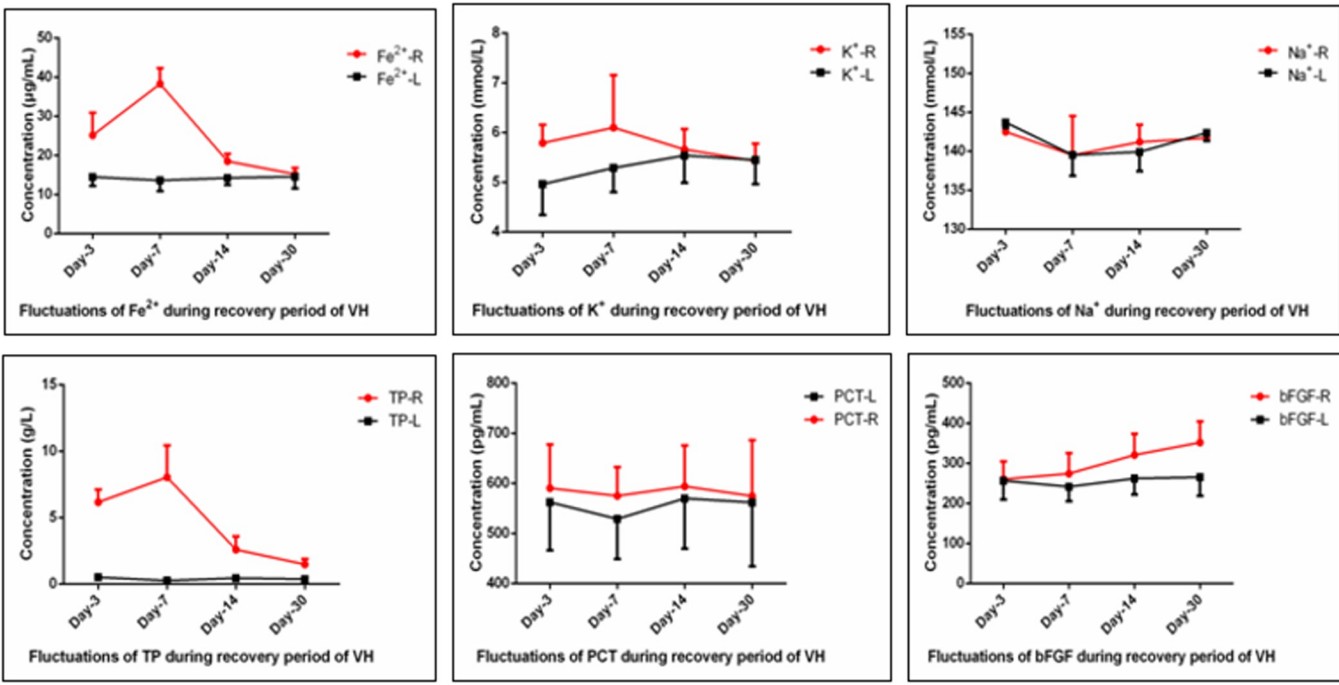

**Fig 2. Line chart of changes in biochemical indicators after VH.**

## Biochemical markers

Fig 2 shows that, unlike the control group (left eyes, L), concentration of iron, potassium, and TP in the study one (right eyes, R) obviously increased on day 3 ($p < 0.05$). All the indicators above peaked on day 7, and then returned to baseline. Moreover, concentration of bFGF in the study group gradually increased (204.30 to 426.00, 301.62±60.90 pg/mL), with the difference beginning to be statistically significant on day 14 ($t = 4.816$, $p = 0.002$). There was a moderate linear correlation between bFGF and degree of liquefaction (pearson $r = 0.658$, $p < 0.001$). No difference in concentrations of sodium and PCT between the two groups (all $p > 0.05$).

## Ultrastructure of VRI

Fig 3 shows that the ultrastructural images of VRI observed under electron microscopes. The binocular response to interventions was analyzed by recording an average number of macrophages and the occurrence of PVD seen within 3 visual fields at the specified spots. SEM showed that PVD appeared as the ILM became smooth and exposed with the lysis of erythrocytes and the sparse and disordered arrangement of collagen fibers. Under the TEM, PVD was induced by cell chemotaxis in the VRI, especially the attachment of activated macrophages. Chi-square test showed that PVD in study group obviously increased from the 14th day ($\chi^2 = 7.385$, $p = 0.007$), and most of them were incomplete PVD, which appeared near the retina (Table 3). In addition, the incidence of PVD was highly positively correlated with the number of macrophages (pearson $r = 0.898$, $p < 0.001$).

## Discussion

This study, which using 2 sets of rabbits, focused on vitreous body, attempted to explore the series of changes after VH and the effects on complications, so as to provide ideas for

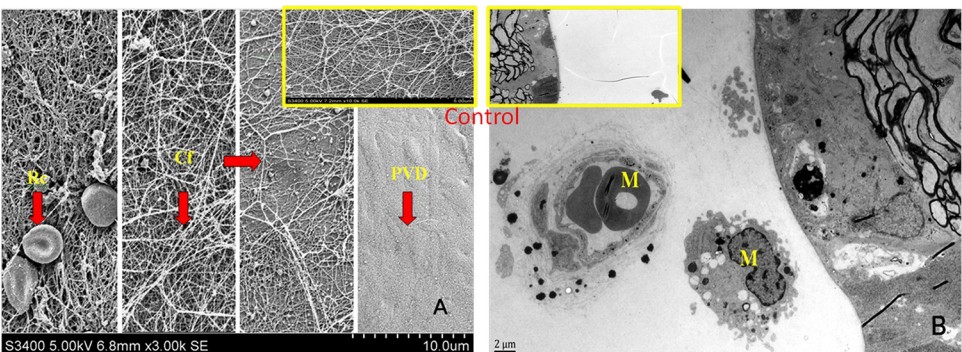

**Fig 3. Ultrastructure of vitreoretinal interface.** (Study eye: A. SEM ×3000, B. TEM ×4000; Control eye). Rc: red blood cells were interwoven with collagen fibers. Cf: the collagen fibers were distributed sparsely. PVD: VRI was damaged irregularly, vitreous cortex peeled from the ILM of retina. M: macrophages attached to the VRI and induced PVD.

investigating the pathogenesis of VH progression. The disclosure of changes in these properties also has certain clinical application values, such as design of ideal vitreous substitute, analysis of intravitreal drug transport, update for prevention strategies, etc. In this study, no obvious behavioral abnormalities were noted, and IOP did not fluctuate significantly. Retina can self-regulate blood supply and discharge under a certain IOP range, but animal studies had shown that retinal detachment is directly related to the degree of vasospasm in the ophthalmic artery during decompensated [18]. Blood injection resulted in rapid coagulum formation, followed by clearance due to fibrinolysis and PVD. It found that the extent of vitreous opacity and coagulum size decreased over time within one month, which exhibited no behavioral and retinal abnormalities. Thus, it is reasonable to wait for spontaneous clearance of VH within 4 weeks.

It is due to its intrinsic biomechanical characteristics, such as liquefaction, viscoelasticity, heterogeneity, and dynamic deformation [19, 20], that allows vitreous body to absorb impacts and protect the retina and lens by acting as a mechanical damper. Changes of such properties can be the origin of later complications. During the test, the liquefaction was aggravated due to the formation of liquefaction pools caused by the separation of water and collagen. As one of the components, collagen contributes to the stiffness of the gel, and the viscosity of vitreous gel decreases with the increase of stiffness after collagen concentration. Fig 1 shows that liquefaction in the study eye started 14 days after blood injection, and accelerated in the control one 30 days postoperatively. With the increasing degree of liquefaction, the viscosity decreases (Table 2), allowing the erythrocytes to diffuse around and facilitate absorption. However, when the integrity of vitreous is broken, a simultaneous increase in liquid volume and

**Table 3. PVD and average number of macrophages according to intervals of time (n = 8).**

| Time intervals | Study group | | | | Control group | | | |
|---|---|---|---|---|---|---|---|---|
| | Near the ciliary body | | Near the retina | | Near the ciliary body | | Near the retina | |
| | PVD | Macrophages | PVD | Macrophages | PVD | Macrophages | PVD | Macrophages |
| Day 3 | 0 | 0 | 1 | 0 | 0 | 0 | 0 | 0 |
| Day 7 | 0 | 0 | 2 | 0 | 0 | 0 | 0 | 0 |
| Day 14 | 1 | 3 | 5 | 8 | 0 | 1 | 0 | 3 |
| Day 30 | 1 | 1 | 6 | 9 | 0 | 0 | 0 | 1 |

stiffening of localized gel could disrupt the balance of stress distribution in eye, placing more stress on the retina potentially [10]. So should slight liquefaction be induced to favor the absorption? The shape of the detached vitreous has been linked to the intensity of vitreoretinal tractions [21]. This study suggests that VH should be treated with medication for approximately 4 weeks and it can be continued if the coagulum sunk to give a clear vision. Otherwise, vitrectomy should be considered to prevent serious complications. Hayashida et al. [22] held that vitrectomy within two weeks may prevent poor visual outcomes, especially in dense VH with unclear etiology. Additionally, liquefaction in the control group was considered to be caused by defensive stress after exposure to intraocular tissue antigens, similar to sympathetic ophthalmia. Therefore, appropriate and prompt therapy for VH is essential.

As can be seen from Fig 2, the biochemical indicators fluctuated temporarily, except for bFGF. It was found that the concentrations of iron, potassium, and TP in the study group obviously increased on day 3 and peaked on day 7, without fluctuations in the concentrations of sodium and PCT. Sodium-potassium ATPase, which can trigger receptor potential, balance osmotic pressure, and regulate solute concentration, is widely distributed in the layers of the retina [23]. Because of dual functions of enzyme and carrier, its activity can be used as an indicator to evaluate retinal cells. Our previous studies confirmed that, the activities of catalase and sodium-potassium ATPase can be drastically reduced by removal of the intact gel, which induces oxidation and aggregation of the lens [11]. The relationship of vitreous composition and biomechanics has been investigated [24]. We propose that a brief increase and redistribution of electrolytes could further disrupt the intact gel, while rapid recovery indicates that attenuation of function of blood-retinal barrier is reversible.

Beside the protection of myelin, there are also various intraocular protective factors, such as bFGF, which has important biological activities in promoting development, proliferation, and differentiation [13]. Our test showed that, the level of vitreous bFGF was moderately correlated with the degree of liquefaction, indicating that bFGF may be involved in the progression of liquefaction and PVD after VH, and even affect the severity of PVR. Li JK, et al. [25] found that the higher level of vitreous bFGF, but not VEGF, correlated with the degree of vitreoretinal fibrosis. However, studies in primates have demonstrated that, RC28-E, as a VEGF/bFGF dual decoy receptor, can be rapidly and evenly distributed into ocular tissues after intravitreal administration, and cross the blood-ocular barrier, which has exhibited high clinical value [26]. Hsuan et al. [27] have reported that bFGF has neurotrophic and protective effects, which could maintain survival, delay the degeneration of neurons, and promote the growth and regeneration of nerve cells. After a pooled analysis, we reasonably speculated that expression of bFGF will be upregulated after VH to induce several kinds of cells to the wound in response to the chemoattractant. If out of control, bFGF can be a double-edged sword, initiating PVR. PVR is an excessive repair response of the retina to trauma, which determines the therapeutic effects [28, 29]. Therefore, determining how these growth factors work is a key to novel therapeutic strategies. Targeting bFGF can effectively neutralize the bioactivity of PVR, and the mass concentration of bFGF in gel is directly proportional to the severity of PVR [30, 31].

After intravitreal injection, erythrocytes lysis, hemoglobin releases iron ions, macrophages release superoxide radicals, and inflammation activates enzymatic reactions, which together lead to vitreous changes. It is the fluid flow from ciliary body to optic disc that initiate the macrophages to clear erythrocytes through loose space around the disc. The results showed an increase in PVD from the 14th day, with more incomplete PVD appearing near the retina induced by activated macrophages. PVD may be related to collagenase or elastase released by macrophages [32]. Notably, given the correlation between bFGF and liquefaction, as well as between macrophages numbers and PVD, it can be speculated that bFGF and macrophages play a vital role in the progression of VH. Based on the ultrastructure of VRI derived from

human donors and animals, it is proposed that the main reason for fibroblast deficiency is barriers-related [33]. Our associated investigators compared the efficacy of combined and sequential surgery for proliferative diabetic retinopathy. They found that the combined surgery had more fibrinous exudation, although both were effective [34]. It suggests that the changes in vitreous body are reversible within 2–4 weeks, which may provide clues for the timing of sequential surgery.

## Conclusions

In summary, an interdisciplinary study for vitreous body was initially attempted, focusing on the effects of hemorrhage on its biomechanics, biochemistry and ultrastructure. After VH, changes of the vitreous are relatively minor earlier (2–4 weeks), but irreversible later. The degree of liquefaction increases with a decrease in viscosity, and chemotaxis of macrophages and bFGF induce incomplete PVD. But, there were also limitations, including the absence of a blank control group, the failure to subdivide the injection volume in the model construction, and imprecision associated with the results due to the infancy interdisciplinary research on vitreous. Hence, further cooperation, focusing more on biomechanics and electrolytes, is urgently needed to achieve the goal of in vivo measurement and analysis for vitreous properties.

## Supporting information

**S1 Checklist. The ARRIVE guidelines 2.0: Author checklist.**
(PDF)

## Acknowledgments

We greatly thanks help of the rabbit house and techniques from Department of Ophthalmology, Tangdu Hospital, Xi'an, China.

## Author Contributions

**Conceptualization:** Pengcheng Zhang, Hong Yan.

**Data curation:** Pengcheng Zhang, Weijia Yan.

**Formal analysis:** Pengcheng Zhang.

**Funding acquisition:** Hong Yan.

**Investigation:** Pengcheng Zhang, Weijia Yan.

**Supervision:** Hong Yan.

**Writing – original draft:** Pengcheng Zhang.

**Writing – review & editing:** Hong Yan.

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
