## [Decision Letter · Decision Letter 0]

19 Sep 2022

PONE-D-22-22747Changes in the Vitreous Body after Experimental Vitreous Hemorrhage in Rabbit: An Interdisciplinary StudyPLOS ONE

Dear Dr. Yan,

Thank you for submitting your manuscript to PLOS ONE. After careful consideration, we feel that it has merit but does not fully meet PLOS ONE’s publication criteria as it currently stands. Therefore, we invite you to submit a revised version of the manuscript that addresses the points raised during the review process.

We look forward to receiving your revised manuscript.

Kind regards,

Prof. Andrzej Grzybowski, MD, PhD, MBA, MAE

Academic Editor

PLOS ONE

Journal Requirements:

2. As part of your revision, please complete and submit a copy of the Full ARRIVE 2.0 Guidelines checklist, a document that aims to improve experimental reporting and reproducibility of animal studies for purposes of post-publication data analysis and reproducibility: https://arriveguidelines.org/sites/arrive/files/documents/Author%20Checklist%20-%20Full.pdf (PDF). Please include your completed checklist as a Supporting Information file. Note that if your paper is accepted for publication, this checklist will be published as part of your article.

"This research was supported by the Tangdu Hospital Grant for Clinical Innovation."

"This research was supported by National Nature Science Foundation of

China (81873674, 82070947, to Hong Yan), Xi’an Fourth Hospital Research Incubation

Fund (LH-6) and Xi’an Talent Program (XAYC200021). The financial sources had no

role in study desigh, data collection and analysis, decision to publish, or preparation of manuscript."

Reviewers' comments:

Reviewer's Responses to Questions

**Comments to the Author**

1. Is the manuscript technically sound, and do the data support the conclusions?

Reviewer #1: Yes

Reviewer #2: Yes

2. Has the statistical analysis been performed appropriately and rigorously? 

Reviewer #1: No

Reviewer #2: Yes

3. Have the authors made all data underlying the findings in their manuscript fully available?

Reviewer #1: Yes

Reviewer #2: Yes

4. Is the manuscript presented in an intelligible fashion and written in standard English?

Reviewer #1: Yes

Reviewer #2: Yes

5. Review Comments to the Author

Reviewer #1: The authors nvestigated the changes in vitreous body after vitreous hemorrhage and assess its prognosis 18 from the perspective of vitreoretinal interface

Currently, medicine has gone through moments of great renewal(1), and advanced methods s have been used(2). Many complications have been observed in the medical practice, such as the vitreous, may also lead to retinal detachment(3).

An optic nerve is an outward form of the diencephalon during embryogenesis, wrapped by a nerve sheath that is derived from three layers of meninges and protrudes toward the orbit(4). As a consequence of this communication, cerebrospinal fluid (CSF) can transfer freely between the intracranial and intraorbital subarachnoid space(5). ICP is transmitted within the optic nerve sheath to obstruct the venous drainage from the eye. Intra-ocular pressure (IOP) was likely elevated after injection blood or serum. Experimentally, a low volumes of intravitreal blood may also cause toxicity in rabbit retina. IOP is regulated primarily through changes in aqueous humor formation and outflow. .

Similarity analyze reveals that the manuscript is similar to other article(without references) ata rate 35%. It should be less than 30%.

There are to aricle which are similar at a rate 8%. One of them is unpublished PDF.

Other is an article (doi: https://doi.org/10.1101/2020.07.09.20150136) but this article was not aslo cited.

The aim of authors is notclear in the introduction section, but they wrote in first sentence of abstract. It should be emphasize in the introduction section.

In order to understand the changes after vitrous hemorrhage, the authors performed experiments in a rabbit model by injection of blood or saline, as well as control animals that did not undergo, so a SHAM group was used, but the finding of this group was not given in the result section

References

1. Gasenzer ERER, Kanat A, Ozdemir V, Neugebauer E. Analyzing of dark past and bright present of neurosurgical history with a picture of musicians. British Journal of Neurosurgery. 2018 May;1–2.

2. Kanat AA, Tsianaka EE, Gasenzer ERE, Drosos EE. Some Interesting Points of Competition of X-Ray using during the Greco-Ottoman War in 1897 and Development of Neurosurgical Radiology: A Reminiscence. Turk Neurosurg. 2021;accepted.

3. Findik H, Kanat A, Aydin MD, Cakir M, Ozmen SA, Okutucu M, et al. Describing a New Mechanism of Retinal Detachment Secondary to Ophthalmic Artery Vasospasm following Subarachnoid Hemorrhage: An Experimental Study. J Neurol Surgery, Part A Cent Eur Neurosurg. 2019 Aug;80(6):430–40.

4. Kazdal H, Kanat A, Findik H, Sen A, Ozdemir B, Batcik OE, et al. Transorbital Ultrasonographic Measurement of Optic Nerve Sheath Diameter for Intracranial Midline Shift in Patients with Head Trauma. World Neurosurg. 2016 Jan;85(1):292–7.

5. Kanat A, Kazdal H, Findik H. In Reply to Letter to the Editor Regarding “Transorbital Ultrasonographic Measurement of Optic Nerve Sheath Diameter for Intracranial Midline Shift in Patients with Head Trauma”. World Neurosurg. 2019 Oct;130:586.

Reviewer #2: This study investigated the changes in vitreous body after vitreous hemorrhage and evaluated its prognosis from the perspective of vitreous interface. The vitreous hemorrhage model in this study is novel, and interdisciplinary research is very meaningful. This experiment designed to explore the changes in the vitreous body after vitreous hemorrhage (VH) in rabbits. They tested IOP, absorption, vitreous retina interface (VRI) and vitreous properties. The results could verify the purposes. Further research should focus more on the biomechanical parameters and electrolytes that significantly changed in the vitreous hemorrhage model.

There are number of minor errors and require the revision:

1. The motivation of study require to add to Introduction and the implication for clinical application also need to provide in Discussion.

2. Line 36 of the article: “and then subsequently…”, the then should be dropped.

Line 85, “fixed” should be replaced with “cured”

3. The English writing needs to improve by professional editor.

6. PLOS authors have the option to publish the peer review history of their article (what does this mean?). If published, this will include your full peer review and any attached files.

Reviewer #1: **Yes: **Ayhan Kanat

Reviewer #2: No

---

## [Author Response · Author response to Decision Letter 0]

21 Oct 2022

Response letter

Dear Reviewers,

On behalf of my co-authors, we are very grateful to you for giving us an opportunity to revise our manuscript. Meanwhile, we appreciate you very much for your positive and constructive comments and suggestions on our manuscript entitled “Changes in the Vitreous Body after Experimental Vitreous Hemorrhage in Rabbit: An Interdisciplinary Study” (ID: PONE-D-22-22747).

We have studied reviewers’ comments carefully and tried our best to revise and improve the manuscript according to the comments. The following are responses and revisions I have made on an item-by-item basis.

Responds to the reviewers’ comments:

Reviewer #1: Dear reviewer Ayhan Kanat,

1) Similarity analyze reveals that the manuscript is similar to other article(without references) ata rate 35%. It should be less than 30%. There are to aricle which are similar at a rate 8%. One of them is unpublished PDF. Other is an article (doi: https://doi.org/10.1101/2020.07.09.20150136) but this article was not aslo cited.

Response: We appreciate it very much for this excellent suggestion, and we have done it according to your comments. As for the large similarity, we have made substantial modifications in sections and sentences, and quoted the literature you mentioned above (doi: https://doi.org/10.1101/2020.07.09.20150136) as the first reference in this manuscript. We do hope that the original degree and readability have been substantially improved.

2) The aim of authors is not clear in the introduction section, but they wrote in first sentence of abstract. It should be emphasize in the introduction section.

Response: We have supplied the aim of this study to the manuscript in the introduction section, line 68-70.

3) In order to understand the changes after vitrous hemorrhage, the authors performed experiments in a rabbit model by injection of blood or saline, as well as control animals that did not undergo, so a SHAM group was used, but the finding of this group was not given in the result section.

Response: In our study, the eyes of rabbits were assigned into two groups: study group (right eyes) – full autologous blood injected, and control one (left eyes, evaluate possible adverse effects of ocular trauma or exogenous fluid) – injected with equal volumes of saline in the same manner (Methods/Animal model construction/Line 98-101). No blank control animals that did not undergo (as a SHAM group) were included and not required to justify our statements. However, the ultrastructure images of the control group (saline injected) were further supplemented. And we will be happy to edit the text further, based on helpful comments from you.

Special thanks to you for your thoughtful comments.

Reviewer #2

1. The motivation of study require to add to Introduction and the implication for clinical application also need to provide in Discussion.

Response: We have supplied the study motivation and clinical application values in the introduction and conclusion sections respectively.

2. Line 36 of the article: “and then subsequently…”, the then should be dropped.

Line 85, “fixed” should be replaced with “cured”

Response: We are very sorry for our negligence of the presentation and the above errors you mentioned have been rectified (Line 59 and 138). Thanks for your advice.

3. The English writing needs to improve by professional editor.

Response: We apologize for some imprecise language of this paper. We worked on it for a long time, while the repeated addition and removal of sentences led to labored readability. We have now worked on both language and readability and have also involved native English speakers for corrections. We do hope that the language level and flow have been substantially improved.

Thanks again to the hard work of you! 

We look forward to receiving your reply.

Kind regards,

Pengcheng Zhang

zhangipengcheng@outlook.com

Corresponding author: Hong Yan

E-mail: yan2128ts@med.nwu.edu.cn

---

## [Decision Letter · Decision Letter 1]

2 Jan 2023

PONE-D-22-22747R1Changes in the vitreous body after experimental vitreous hemorrhage in rabbit: an interdisciplinary studyPLOS ONE

Dear Dr. Yan,

Thank you for submitting your manuscript to PLOS ONE. After careful consideration, we feel that it has merit but does not fully meet PLOS ONE’s publication criteria as it currently stands. Therefore, we invite you to submit a revised version of the manuscript that addresses the points raised during the review process.

We look forward to receiving your revised manuscript.

Kind regards,

Andrzej Grzybowski

Academic Editor

PLOS ONE

Journal Requirements:

Reviewers' comments:

Reviewer's Responses to Questions

**Comments to the Author**

1. If the authors have adequately addressed your comments raised in a previous round of review and you feel that this manuscript is now acceptable for publication, you may indicate that here to bypass the “Comments to the Author” section, enter your conflict of interest statement in the “Confidential to Editor” section, and submit your "Accept" recommendation.

Reviewer #1: All comments have been addressed

Reviewer #2: All comments have been addressed

2. Is the manuscript technically sound, and do the data support the conclusions?

Reviewer #1: Partly

Reviewer #2: Yes

3. Has the statistical analysis been performed appropriately and rigorously? 

Reviewer #1: Yes

Reviewer #2: Yes

4. Have the authors made all data underlying the findings in their manuscript fully available?

Reviewer #1: No

Reviewer #2: Yes

5. Is the manuscript presented in an intelligible fashion and written in standard English?

Reviewer #1: No

Reviewer #2: Yes

6. Review Comments to the Author

Reviewer #1: This interdisciplinary study focused on the effects of hemorrhage on biomechanics, biochemistry and ultrastructure of vitreous body. Vitreous hemorrhage is an important issue, because it may lead to persistent loss of vision(1). In this study, the eyes of rabbits were assigned into two groups: study group (right eyes) – full autologous blood injected, and control one (left eyes, evaluate possible adverse effects of ocular trauma or exogenous fluid) – injected with equal volumes of saline in the same manner. No blank control animals that did not undergo.

The author agrre that a control group is not required to justify their statements.

Is there any previous study that use this method?

If here is, I recommend them to cite that study.

If not, this situation should be noted in the limitation section.

The ultrastructure images of the control group (saline injected) were supplemented. It is difficult to accept the saline injected animal as a control group animal.

As part of the central nervous system, the optic nerve is surrounded by a distensible subarachnoidal space, designated as the optic nerve sheath(2). As a consequence of this communication, CSF can transfer freely between the intracranial and intraorbital subarachnoid space(3). This communication may be the reason why papilledema be a common neuro-opthalmological exam finding of intraocular pressure elevation. In this study, after intravitreal injection of blood or saline , IOP did not fluctuate significantly. How can the authors explain this situation?

Findik et al found that retinal detachment is directly related to the degree of vasospasm in the ophthalmic artery(1). I have a concern about the author’ suggestion about that the retina can dynamically maintain the blood flow rate constant. What do you think the authors about the vasospasm in the ophthalmic artery? The manuscript require minor revision

References

1. Findik H, Kanat A, Aydin MD, Cakir M, Ozmen SA, Okutucu M, et al. Describing a New Mechanism of Retinal Detachment Secondary to Ophthalmic Artery Vasospasm following Subarachnoid Hemorrhage: An Experimental Study. J Neurol Surgery, Part A Cent Eur Neurosurg. 2019 Aug;80(6):430–40.

2. Guvercin AR, Besir A, Kanat A, Yazar U, Findik H. Interesting Negative Correlation between Transorbital Optic Nerve Sheath Diameter and Evans’ index Values; Can it Be Predictive for Failure of Endoscopic Third Ventriculostomy? Int J Neurosci. 2022 Sep;1–11.

3. Kazdal H, Kanat A, Findik H, Sen A, Ozdemir B, Batcik OEOE, et al. Transorbital Ultrasonographic Measurement of Optic Nerve Sheath Diameter for Intracranial Midline Shift in Patients with Head Trauma. World Neurosurg. 2016 Jan;85(1):292–7.

Reviewer #2: All comments have been addressed. This manuscript is technically sound, and the data support the conclusions.

7. PLOS authors have the option to publish the peer review history of their article (what does this mean?). If published, this will include your full peer review and any attached files.

Reviewer #1: **Yes: **Ayhan Kanat

Reviewer #2: No

---

## [Author Response · Author response to Decision Letter 1]

13 Jan 2023

Response letter

Dear Reviewers,

On behalf of my co-authors, we are very grateful to you for your positive and constructive comments and suggestions on our manuscript entitled “Changes in the Vitreous Body after Experimental Vitreous Hemorrhage in Rabbit: An Interdisciplinary Study” [PONE-D-22-22747R2] - [EMID:869453ce1fd5a818].

We have studied the comments carefully and tried our best to revise and improve the manuscript according to the comments. The following are responses and revisions we have made on an item-by-item basis.

Responds to the reviewers’ comments:

Reviewer #1: Dear professor Ayhan Kanat,

1) The author agrre that a control group is not required to justify their statements. Is there any previous study that use this method? If here is, I recommend them to cite that study. If not, this situation should be noted in the limitation section. The ultrastructure images of the control group (saline injected) were supplemented. It is difficult to accept the saline injected animal as a control group animal.

Response: The intervention used in this study was an intravitreal injection of autologous blood, to explore the changes of vitreous itself after hemorrhage. That is, the focus is on intravitreous hemorrhage, rather than intravitreous injection. Therefore, a blank control in this study was not selected, but saline intravitreal injection. Similar method was also used for comparison, which has been cited in the manuscript (Line 101, new citations 15 and 16). Of course, it would be better to include blank control as well. Hence, it is pointed out in the limitation section (Line 305-306) that needs to be improved. We appreciate it very much for this excellent suggestion, and we will be happy to edit the text further, based on helpful comments from you.

2) As part of the central nervous system, the optic nerve is surrounded by a distensible subarachnoidal space, designated as the optic nerve sheath. As a consequence of this communication, CSF can transfer freely between the intracranial and intraorbital subarachnoid space. This communication may be the reason why papilledema be a common neuro-opthalmological exam finding of intraocular pressure elevation. In this study, after intravitreal injection of blood or saline , IOP did not fluctuate significantly. How can the authors explain this situation?

Response: It is true that the optic nerve is sensitive to changes in intraocular pressure. Vitreous fluid injection also likely causes IOP elevation. Thus, in the “Animal model construction” section of this manuscript, we mentioned that after the injection, the “anterior chamber puncture” procedure (Line 104-105) is designed to soften the eyeball, precisely to avoid excessive fluctuations of intraocular pressure.

3) I have a concern about the author’ suggestion about that the retina can dynamically maintain the blood flow rate constant. What do you think the authors about the vasospasm in the ophthalmic artery? The manuscript require minor revision.

Response: Special thanks to you for this valuable feedback. The dynamic regulation of retina for blood flow rate exists under a certain IOP range in physiological conditions. Once decompensation occurs, vasospasm or lesions may occur, that retinal detachment is directly related to the degree of vasospasm in the ophthalmic artery. We have revised the relevant content (Line 225-227, newly cited reference 19). 

Reviewer #2: Dear Reviewer,

Thank you for appreciating our efforts to address your constructive comments, which really helped us improve the manuscript.

Thank both reviewers for your time and positive feedback for the evaluation of the revision.

We look forward to receiving your reply.

Kind regards,

Pengcheng Zhang

zhangipengcheng@outlook.com

Corresponding author: Hong Yan

E-mail: yan2128ts@med.nwu.edu.cn

---

## [Editor Report · Decision Letter 2]

17 Jan 2023

Changes in the vitreous body after experimental vitreous hemorrhage in rabbit: an interdisciplinary study

PONE-D-22-22747R2

Dear Dr. Yan,

We’re pleased to inform you that your manuscript has been judged scientifically suitable for publication and will be formally accepted for publication once it meets all outstanding technical requirements.

Kind regards,

Andrzej Grzybowski

Academic Editor

PLOS ONE
---

## [Editor Report · Acceptance letter]

24 Jan 2023

PONE-D-22-22747R2 

Changes in the vitreous body after experimental vitreous hemorrhage in rabbit: an interdisciplinary study 

Dear Dr. Yan:

I'm pleased to inform you that your manuscript has been deemed suitable for publication in PLOS ONE. Congratulations! Your manuscript is now with our production department. 

Kind regards, 

on behalf of

Dr. Andrzej Grzybowski 

Academic Editor

PLOS ONE